# A time transect of exomes from a Native American population before and after European contact

John Lindo[1], Emilia Huerta-Sánchez[2], Shigeki Nakagome[1], Morten Rasmussen[3,4], Barbara Petzelt[5], Joycelynn Mitchell[5], Jerome S. Cybulski[6,7,8], Eske Willerslev[3,9,10], Michael DeGiorgio[11,12] & Ripan S. Malhi[13,14]

A major factor for the population decline of Native Americans after European contact has been attributed to infectious disease susceptibility. To investigate whether a pre-existing genetic component contributed to this phenomenon, here we analyse 50 exomes of a continuous population from the Northwest Coast of North America, dating from before and after European contact. We model the population collapse after European contact, inferring a 57% reduction in effective population size. We also identify signatures of positive selection on immune-related genes in the ancient but not the modern group, with the strongest signal deriving from the human leucocyte antigen (HLA) gene *HLA-DQA1*. The modern individuals show a marked frequency decrease in the same alleles, likely due to the environmental change associated with European colonization, whereby negative selection may have acted on the same gene after contact. The evident shift in selection pressures correlates to the regional European-borne epidemics of the 1800s.

[1] Department of Human Genetics, University of Chicago, 920 E 58th Street, Chicago, Illinois 60637, USA. [2] Department of Molecular Cell Biology, University of California, 5200 North Lake Road, Merced, California 95343, USA. [3] Centre for GeoGenetics, Natural History Museum of Denmark, University of Copenhagen, Øster Voldgade 5–7, DK-1350 Copenhagen K, Denmark. [4] Department of Genetics, School of Medicine, Stanford University, 291 Campus Drive, Stanford, California 94305, USA. [5] Metlakatla Treaty Office, PO Box 224, Prince Rupert, British Columbia V8J 3P6, Canada. [6] Canadian Museum of History, 100 Rue Laurier, Gatineau, Quebec K1A 0M8, Canada. [7] Department of Anthropology, University of Western Ontario, London, Ontario N6A 3K7, Canada. [8] Department of Archaeology, Simon Fraser University, Burnaby, British Columbia V5A 1S6, Canada. [9] Department of Zoology, University of Cambridge, Downing St., Cambridge CB2 3EJ, UK. [10] Wellcome Trust Sanger Institute, Wellcome Genome Campus, Hinxton, Cambridge Cb10 1SA, UK. [11] Departments of Biology and Statistics, Pennsylvania State University, 502 Wartik Laboratory, University Park, Pennsylvania 16802, USA. [12] Institute for CyberScience, Pennsylvania State University, 502 Wartik Laboratory, University Park, Pennsylvania 16802, USA. [13] Carl R. Woese Institute for Genomic Biology, University of Illinois, 1206 W Gregory Dr., Urbana, Illinois 61820, USA. [14] Department of Anthropology, University of Illinois, 607 S. Mathews Ave., Urbana, Illinois 61820, USA. Correspondence and requests for materials should be addressed to M.D. (email: mxd60@psu.edu) or to R.S.M. (email: malhi@illinois.edu).

The decline of Native American populations after European contact has been linked to several factors including warfare, alterations in social structure and an overwhelming introduction of European-borne pathogens[1–3]. Although the extent of the population decline remains contentious, European-borne epidemics may have disproportionately contributed to the phenomenon[4,5]. The debate has prompted researchers to explore the possibility of genetic susceptibility, where low-genetic variation in HLA genes and immunologically naïve populations are linked to the exacerbated pathogen-associated mortality rates[6–8]. Assumptions of homogeneity among certain immune genes[8], however, are based on surveys of living Native Americans who represent the surviving members of communities affected by European contact and colonization. Thus, they fail to consider immune-related genetic factors that may have existed before contact.

The immunological history of the indigenous people of the Americas is undoubtedly complex. As people entered the Americas and expanded into different regions, ∼15,000–20,000 years before present (BP)[9,10], groups encountered environments with varying ecologies and with relatively little gene flow from other continental populations until European contact[11]. We hypothesize that indigenous people adapted to local pathogens, resulting in long-lasting changes to immune-related loci. Ancient immune adaptations are suspected to have occurred throughout human history as populations spread into varying environments across the globe[12]. If the indigenous people of America adapted to local pathogens, those adaptations would have proven useful in ancient times but not necessarily after European colonialists altered the environment with their pathogens, some of which may have been novel[13–15]. Existing genetic variation as a result of adaptation before European contact could thus have contributed to the indigenous population decline after European contact.

To investigate the possibility of a pre-existing genetic component, we sequenced 50 exomes of ancient and modern individuals from the Northwest Coast of North America, dating from before and after European contact. We confirm the genetic continuity between the ancient and modern individuals, establishing a single continuous population through time. We show a 57% reduction in effective population size after European contact, inferred from our demographic model. We also detect signatures of positive selection on immune-related genes in the ancient but not the modern individuals. The strongest selection signal in the ancients derives from the human leucocyte antigen (HLA) gene HLA-DQA1, with alleles that are close to fixation. The important immune function of HLA-DQA1 supports an ancient adaptation to the environments of the Americas. The modern individuals show a marked decrease in the frequency of the associated alleles (the most pronounced variant showing a 64% difference). This decrease is likely due to the environmental change associated with European colonization, which resulted in a shift of selection pressures, whereby negative selection may have acted on the same gene after contact. Furthermore, the selection pressure shift could correlate to the European-borne epidemics of the 1800s, suffered in the Northwest Coast region. This is among the first studies to examine a single population through time and exemplifies the power of such studies in uncovering nuanced demographic and adaptive histories.

## Results

**Samples and sequencing.** To investigate possible immune-related genes under selection before European contact, we sequenced the exomes of ancient and modern First Nations individuals of the Prince Rupert Harbour (PRH) region of British Columbia, Canada (Supplementary Fig. 1, Supplementary Note 1). We then performed genomic scans for positive selection and functional characterization of genes exhibiting the strongest signals. Exomes of 25 modern individuals from two Coast Tsimshian communities, Metlakatla and Lax Kw'alaams (henceforth referred to as 'Tsimshian'), were sequenced to a mean depth of $9.66 \times$. The 25 ancient individuals from archaeological sites in the PRH region (henceforth referred to as 'PRH Ancients'; Supplementary Fig. 1) range in age from ∼6,260 to 1,036 cal BP (Supplementary Table 1), with most of the individuals falling between 3,000 and 1,036 cal BP. The ancient exomes were sequenced to a mean depth of $7.97 \times$ (Supplementary Table 2). Contamination estimates (that is, exogenous DNA stemming from modern sources), using the exome-wide data, revealed a mean contamination of 0.94% with a 95% confidence interval of 0.83–1.10% (ref. 16; Supplementary Table 3). All 25 ancient individuals exhibited patterns of $C \rightarrow T$ and $G \rightarrow A$ transitions consistent with deamination due to post-mortem DNA damage[17,18] (Supplementary Fig. 2). Mitochondrial haplogroups were determined for each ancient individual, all showing haplogroups previously identified in Native Americans[19] (Supplementary Table 1).

**Genetic relationship between ancient and modern individuals.** Before proceeding with selection scans we investigated the genetic relationship among the ancient and modern individuals to confirm continuity between the two groups. For these analyses, all C/T and G/A polymorphic sites were removed to guard against biases resulting from DNA damage. Multidimensional scaling was performed to assess the genetic relationships of our samples to individuals from the 1,000 Genomes Project[20], other Native American populations[21] and two ancient individuals from America[22,23] (Fig. 1b, Supplementary Note 2). The analysis revealed an affinity among the PRH Ancients and the Tsimshian, with the Tsimshian drifting towards Europeans as expected from presumed European admixture[24]. We next used ADMIXTURE[25] to separate our samples and other worldwide populations into clusters. This analysis (at $K = 5$ clusters) suggests that the Tsimshian are a mixture of ancestral components stemming from the PRH Ancients and Europeans (Fig. 1a). Next, the evolutionary relationship among our samples and other worldwide human populations was evaluated via TreeMix[26]. With a single-migration event, the PRH Ancients exhibit minimal drift and appear ancestral to the Tsimshian with European admixture occurring between the two groups (Fig. 1c). These analyses, combined with local oral histories and evidence from archaeology and mitogenomes[19], allowed for the inference that the ancient and modern individuals represent a single population through time, which includes pre- and post-European contact.

**Demographic model.** We also inferred the population history of the Tsimshian by taking into account the bottleneck that occurred after European contact[1,27,28] (Fig. 2, see 'Methods' section). Utilizing both ancient and modern exome-wide data, the demographic parameters were inferred utilizing the joint derived site frequency spectrum of potential synonymous sites with respect to human reference genome hg19. The best-fitting model suggests that a bottleneck occurred ∼175 years BP (bootstrap 95% confidence interval: 125–225 years, Table 1) in the ancestors of the modern Tsimshian with an accompanying reduction in effective population size of 57%. The timing of the bottleneck coincides with the documented smallpox epidemics of the 19th Century and historical reports of large-scale population declines[29,30]. A majority of European admixture in the population likely occurred after the epidemics[24,29].

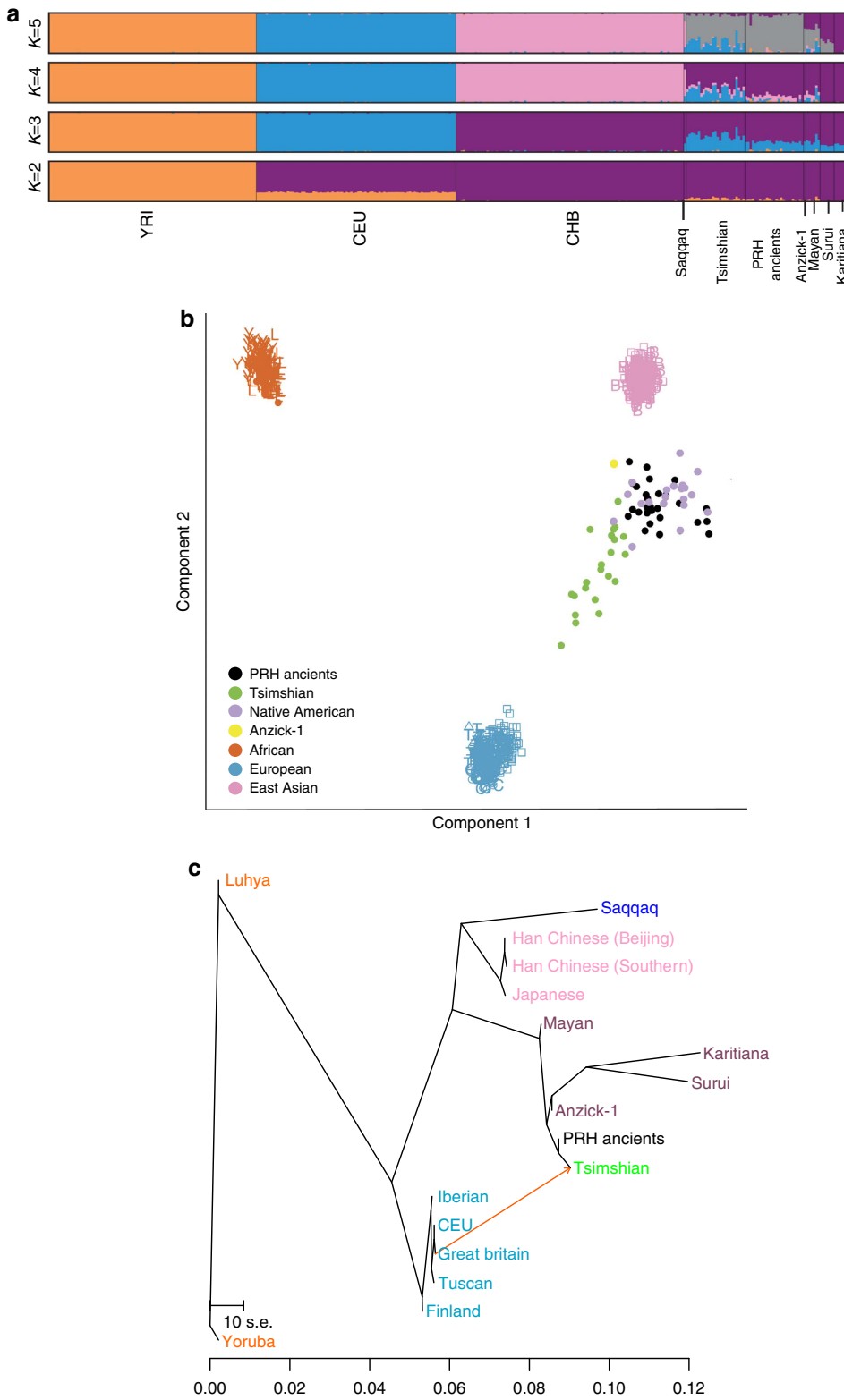

**Figure 1 | Population structure and demographic model of the PRH Ancients and Tsimshian.** (**a**) *ADMIXTURE* analysis depicting ancestry proportions assuming the number of genetic components, *K*, is 2–5. The analyses included reference populations from the 1,000 Genomes Project Phase 2 data set[20], Native American individuals sampled from the Karitiana, Surui and Maya[21], and two ancient samples from the Americas—the Saqqaq[22] and Anzick-1 (ref. 23). (**b**) Multidimensional scaling plot. The Native American (Surui, Mayan and Karitiana) populations fall with the PRH Ancients and the Anzick-1 ancient sample from Montana. The modern Tsimshian fall along the gradient leading from the Native Americans and the Europeans, reflecting their admixed history with Europeans. (**c**) *TreeMix* graph with a single admixture event.

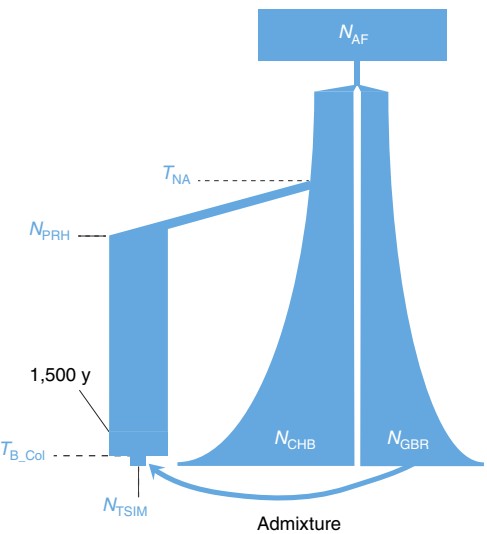

**Figure 2 | Demographic model.** Best-fitting model depicting the Tsimshian bottleneck after European contact and the subsequent admixture with Europeans. Fixed demographic parameters are from Gravel et al.[55] and admixture parameters are from Verdu et al.[24]. The parameters in blue were inferred with FastSimCoal2 (ref. 54). Population sizes and time splits are not shown to scale. The model's labels are defined as: $N_{AF}$, African effective population size ($N_e$); $N_{CHB}$, CHB $N_e$; $N_{GBR}$, GBR $N_e$; $N_{PRH}$, PRH Ancients $N_e$; $N_{Tsim}$, Tsimshian $N_e$; $T_{NA}$, time of split between PRH Ancients and CHB; $T_{B\_Col}$, time of colonial contact bottleneck.

**Table 1 | Parameters estimated for the model displayed in Fig. 2 using FastSimCoal2 (ref. 54).**

| Parameter | Inferred Value | 95% confidence interval |
|---|---|---|
| $N_{PRH}$ | 13,975 | 7,532–15,178 |
| $N_{TSIM}$ | 6,006 | 4,124–8,048 |
| $T_{NA}$ | 15,125 | 14,105–16,870 |
| $T_{B\_Col}$ | 175 | 125–225 |

$N_{PRH}$, $N_e$ PRH Ancients; $N_{TSIM}$, $N_e$ Tsimshian; PRH, Prince Rupert Harbour; $T_{B\_Col}$, time of European contact bottleneck; $T_{NA}$, time of CHB/PRH Ancients split. $N_e$ = effective population size. Population size estimates are for diploid individuals. Time estimates are in years, assuming a generation time of 25 years.

**Scans for positive selection**. To safeguard against false-positive signals of positive selection due to the apparent admixture of Tsimshian individuals with Europeans, we performed an admixture correction (see 'Methods' section). The populations were scanned for selection signals, with and without correcting for admixture, utilizing the population branch statistic (PBS)[31]. PBS has proven effective in detecting positively-selected loci among high-altitude populations[31,32]. Twenty-five Han individuals from Beijing (CHB), part of the 1,000 Genomes Project[20], served as the third comparative population. The statistic computes the amount of differentiation at a given locus along a branch leading to a specific population by comparing transformed $F_{ST}$ values between each pair of three populations. Figure 3a displays population-specific differentiation for the mean across the exome and for our top candidate gene (HLA-DQA1) discussed below.

We calculated PBS on a per-gene basis, with $P$ values for each gene computed by comparing the observed PBS scores with the distribution calculated under neutral simulations. We report genes with $P$ values below the 0.05 significance level (Table 2). The genes showing the most extreme and significant PBS values in the PRH Ancients represent strong candidates for positive selection, of which the top candidate, HLA-DQA1, is directly involved with immune function (Table 2). Enriched gene ontologies were also identified from the ranked list of genes generated from the PBS scan, which highlight immune function related to antigen presentation (Table 3). To assess whether the selection signals were extreme relative to expectations under neutrality, the PBS scores were compared with the distribution of scores based on neutral simulations using our inferred demographic history (Supplementary Fig. 3). Variants from the top candidate with the most pronounced frequency changes were confirmed via Sanger sequencing in all ancient samples reporting data (Supplementary Fig. 4, Supplementary Note 3).

**Relevance of the HLA-DQA1 gene**. The most extreme PBS score belonged to the HLA-DQA1 gene, which encodes for the alpha chain of the major histocompatibility complex (MHC), class II, DQ1 isoform. The HLA-DQA1 single nucleotide polymorphism (SNP) with the most pronounced frequency difference between the PRH Ancients (100%) and the Tsimshian (36%) falls in the 5′ untranslated region (Table 4). This region may be indicative of selection acting on the regulation of the gene, as the associated alleles exhibit evidence of chromatin alterations and eQTL hits in a variety of cells—including monocytes-CD14 + and primary T helper 17 cells[33] (Supplementary Table 4). The chromosomal region where the gene is located also shows strong differentiation along the branch leading to the PRH Ancients (Fig. 3).

HLA-DQ is one of the three main types of MHC class II molecules, along with DR and DP, and is mainly expressed on antigen presenting cells[34]. MHC class II molecules are responsible for binding to extracellular pathogen peptides and presenting them to CD4 + T helper cells, which activate a targeted adaptive immune response towards the associated microbe[35]. The molecules are known to be highly polymorphic, mainly due to sequence differences corresponding to the binding domain of the molecule, which can impact binding affinities[36]. Because of this variety in binding domains, differing MHC class II isoforms can have differing disease outcomes due to the restriction imposed on T-cell activation[35]. The polymorphic nature of these molecules across different populations, however, would not explain the heightened differentiation in the PRH Ancients with respect to their presumed descendants, the Tsimshian.

**Haplotype structure and local ancestry of HLA-DQA1**. The top candidate for selection in the ancient population, HLA-DQA1, showed large-allele frequency changes in the UTR5 region of the modern population (Table 4). Although there is a slight reference bias in the ancient samples due to mapping and possibly the design of our capture probes (Supplementary Fig. 5, Supplementary Note 4), the high frequency cannot be attributed to this feature since the derived alleles putatively under selection are for the alternate allele. To assess whether the frequency change was due to European admixture, we examined the haplotype structure among populations. To visualize the haplotypes in the HLA-DQA1 region, we phased the ancient and modern samples using Beagle 4.1 (ref. 37). We took a randomly chosen haplotype from ancient sample PRH 125, and computed the number of pairwise differences to this haplotype for each haplotype in the modern and ancient samples as well as the Great Britain (GBR) samples from the 1,000 Genomes Phase 3 data[38]. We then ordered the haplotypes based on their number of pairwise differences to this arbitrarily chosen haplotype from sample PRH 125, and grouped them by population. Supplementary Figure 6 shows similar haplotypes between the

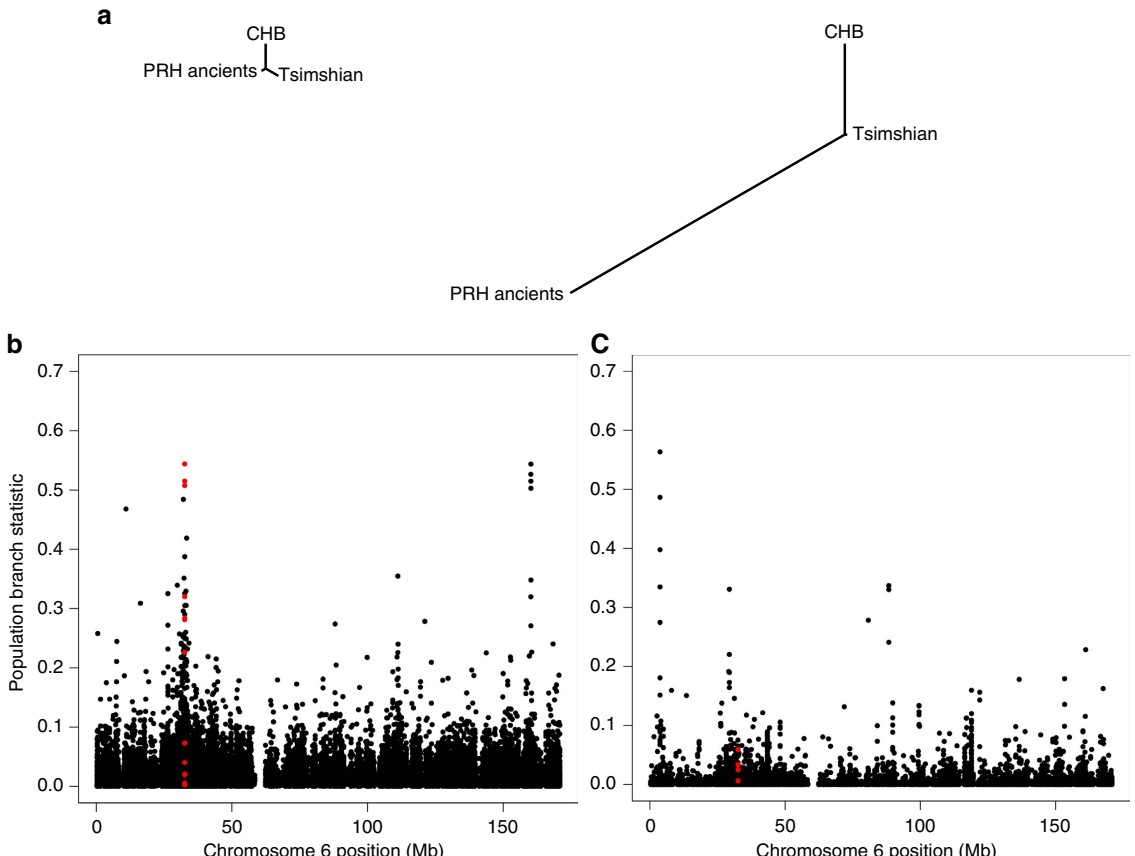

**Figure 3 | Population branch statistic (PBS) of the *HLA-DQA1* gene.** (**a**) Trees based on exome-wide data, with the left tree showing a small branch length of the PRH Ancients relative to Tsimshian and CHB and the right tree showing strong differentiation along the PRH Ancients branch at the *HLA-DQA1* gene. (**b,c**) Manhattan plots of the PBS score calculated on a per-SNP basis as a function of SNP position on chromosome 6 for the PRH Ancients (**b**) and the modern Tsimshian (**c**). The region highlighted in red shows PBS scores for SNPs within 10 kb of the *HLA-DQA1* gene.

**Table 2 | Genes with the strongest frequency changes in the PRH Ancient individuals.**

| Gene | PBS | P value | Sites | Description |
|---|---|---|---|---|
| *HLA-DQA1* | 0.275500845 | 0.01675 | 17 | Major histocompatibility complex, class II, DQ alpha 1 |
| *CELA3A* | 0.194966437 | 0.03102 | 29 | Basic leucine zipper nuclear factor 1 |

PBS, population branch statistic; PRH, Prince Rupert Harbour. *P* values were calculated using neutral simulations. See 'Methods' section.

**Table 3 | Enriched gene ontologies for the PRH Ancient individuals derived from the PBS selection scan ranked list.**

| Ontology enrichment term | P value | FDR q-value |
|---|---|---|
| Peptide antigen binding | $2.61 \times 10^{-8}$ | $1.09 \times 10^{-4}$ |
| MHC class II receptor activity | $8.11 \times 10^{-8}$ | $1.69 \times 10^{-4}$ |
| G-protein coupled receptor activity | $6.12 \times 10^{-6}$ | $8.53 \times 10^{-3}$ |
| Olfactory receptor activity | $7.91 \times 10^{-6}$ | $8.26 \times 10^{-3}$ |
| Antigen binding | $1.36 \times 10^{-5}$ | $1.14 \times 10^{-2}$ |

FDR, false discovery rate; PBS, population branch statistic; PRH, Prince Rupert Harbour. Enrichment terms and significance values calculated via GOrilla[57].

ancient and modern individuals, while those of the European population are distinct.

We next explored the local ancestry of the *HLA-DQA1* gene. We used RFMix[39] to infer ancestry along chromosome 6 in the modern Tsimshian population. We utilized the PopPhased program, which corrects the phasing errors, and a window size of 0.2 cM, four generations since the admixture event between the Tsimshian and Europeans, and 100 trees generated per random forest. For the reference panel, we used Phase 3 data from the 1,000 Genomes Project[38]. We used 25 individuals each from the GBR (European panel), CHB (East Asian panel) and PEL (Native American Panel, Peruvian in Lima). The PEL chosen showed little to no admixture (see 'Methods' section). Supplementary Figure 7 indicates that only one haplotype could be attributed to European ancestry, while the remaining 49 are attributed to Native American ancestry.

**Simulations of the *HLA-DQA1* allele trajectories.** To explore whether the allele frequency differences between the two time periods could be explained by long-term balancing selection, drift or changes in selection pressures, we performed a series of simulations based on our demographic model. First, we examined whether long-term balancing selection under heterozygote advantage could explain our data. The parameters inferred from our demographic model were implemented in the forward-time simulator SLiM[40]. A *de novo* mutation was introduced 5 million years in the past (assuming a generation time of 25 years) that evolved under heterozygote advantage (per-generation selection coefficient $s = 0.1$, and dominance parameter $h = 100$) until the present. The distribution of the resulting PBS scores can be contrasted with the observed data in Fig. 4a. Because the

**Table 4 | Population frequencies for the *HLA-DQA1* SNPs on chromosome 6.**

| Gene segment | SNP ID | Ancestral allele | Derived allele | Exonic function | PRH Ancients frequency | Tsimshian frequency | CHB frequency |
|---|---|---|---|---|---|---|---|
| UTR5 | rs9272426 | A | G | NA | 1 | 0.37 | 0.46 |
| UTR5 | rs3207966 | G | A | NA | 1 | 0.41 | 0.16 |
| UTR5 | rs3187964 | G | C | NA | 0 | 0.55 | 0.49 |
| UTR5 | rs1047985 | G | A | NA | 1 | 0.43 | 0.11 |
| Exonic | rs1047989 | C | A | Nonsynonymous | 1 | 0.55 | 0.51 |
| Exonic | rs1047993 | C | T | Synonymous | 1 | 0.42 | 0.12 |
| Exonic | rs10093 | G | C | Synonymous | 0.89 | 0.67 | 0.79 |
| Exonic | rs1129765 | T | C | Synonymous | 0.84 | 1 | 1 |
| Exonic | rs1142323 | A | G | Nonsynonymous | 0.77 | 0.27 | 0.14 |

PRH, Prince Rupert Harbour; SNP ID, single nucleotide polymorphism ID. The population frequencies are for the derived allele with respect to the chimpanzee reference.

distribution of the PBS scores under long-term balancing selection is shifted towards small values compared with neutrality (Fig. 4a), the data are inconsistent with long-term balancing selection under heterozygote advantage.

To model evolutionary forces acting on the *HLA-DQA1* derived alleles after European contact, we chose the frequency of the allele showing the greatest change of 0.67. We utilized a simulation based approach, described in detail in the 'Methods' section, to evaluate models under positive selection, neutrality and negative selection. We also used the same approach to obtain estimates of the correlation between the time of environmental change ($t$) and the selection coefficient ($s$) (Supplementary Fig. 8). Figure 4c shows that neither a strict positive selection scheme, nor one involving positive selection followed by a shift to neutrality, could fit our data (none of the simulations reach the observed frequency in the modern population). However, the model with a shift from positive to negative selection was compatible, where 26% of the simulations either reach or surpass the observed frequency.

We also investigated if the observed allele frequency in the ancient population could be better explained by drift rather than selection. Using the same general method, we simulated the initial allele frequencies at the time Native Americans split off from East Asian population by randomly sampling allele frequencies from a backward simulation conditioned on the modern CHB frequency. We then simulated allele frequencies from 60 generations ago—the time during which the ancient population was sampled. The resulting distribution in Supplementary Fig. 9 shows that a neutral scenario is not a good fit for our data. We also see that the empirical distribution of all SNP frequency changes between the ancient and modern individuals show the *HLA-DQA1* variants as outliers (Fig. 4b).

## Discussion

Our unique data set has allowed us to examine the demography of a single Native American population through three distinct time frames. We first examined the population from a time span of 5,000 years leading to European contact. Selection scans on the ancient individuals from this period revealed a top candidate for positive selection, *HLA-DQA1*, giving the inference of an immune-related adaptive event. We next inferred the severity of the population collapse after European contact, which correlates with historical population declines associated with regional smallpox epidemics[30], as well as general estimates of Native American population declines based on mitochondrial DNA diversity[27,41]. During the contact period, previous long-standing positive selection on the *HLA-DQA1* gene may also have been significant. The HLA-DQ receptor has been associated with a variety of colonization era infectious disease, including measles[42],

tuberculosis[43,44], and with the adaptive immune response to the vaccinia virus, which is an attenuated form of smallpox[45,46]. Further studies are needed to investigate if the ancient alleles putatively under positive selection may pose a differential disease outcome with respect to European-borne pathogens, as well as their effect on downstream target genes.

However, when examining the population post-contact and into contemporary times, variants of the *HLA-DQA1* gene experience a marked frequency change. This change presents a more complex scenario when taking into account all three time frames. First, scans for positive selection in the modern Tsimshian, with and without correcting for European admixture, revealed no statistically significant selection on immune-related genes (Supplementary Tables 5 and 6). The gene ontology enrichment analyses also did not suggest a correlation with immune function (Supplementary Table 7). Second, demography alone was unable to explain the large frequency change in the *HLA-DQA1* alleles between the ancient and modern groups based on simulations (Fig. 4c). European admixture in the modern individuals also did not account for the frequency changes since the haplotypes in this region can be attributed to Native American ancestry (Supplementary Fig. 7). Furthermore, *HLA-DQA1* remained a top PBS hit in scans involving both a European admixture correction (Supplementary Table 9) and with an additional scan involving unadmixed Native American individuals from a different modern population (suggesting a regional adaptive event) (Supplementary Table 8; ranked fourth best candidate, with the top three functionally uncharacterized).

We therefore explored alternative explanations for the observed frequency change of the *HLA-DQA1* alleles in the time after contact. Since HLA genes have been previously postulated to be under balancing selection in humans[47,48], we examined the possibility that long-term balancing selection could explain our data by simulating under a model of heterozygote advantage conditional on our inferred demographic model. We found that this specific type of balancing selection is a poor fit to the data, whereby the *HLA-DQA1* gene is still an extreme outlier relative to the simulation results (Fig. 4a). Next, we used a forward simulation based approach to trace the *HLA-DQA1* allele trajectories under different selection models after the point of European contact. We found that simulations under our demographic model, which was modified to not include European admixture given our local ancestry results (Supplementary Fig. 7), was insufficient to explain the frequency change in the modern population—with none of the $10^4$ simulations reaching the observed frequency (Fig. 4c). However, on applying a model of negative selection at the time of contact, we found that simulated allele frequencies were compatible with the observed frequencies in the modern population (Fig. 4c). Although we were unable to precisely identify the selection coefficient necessary to drive the

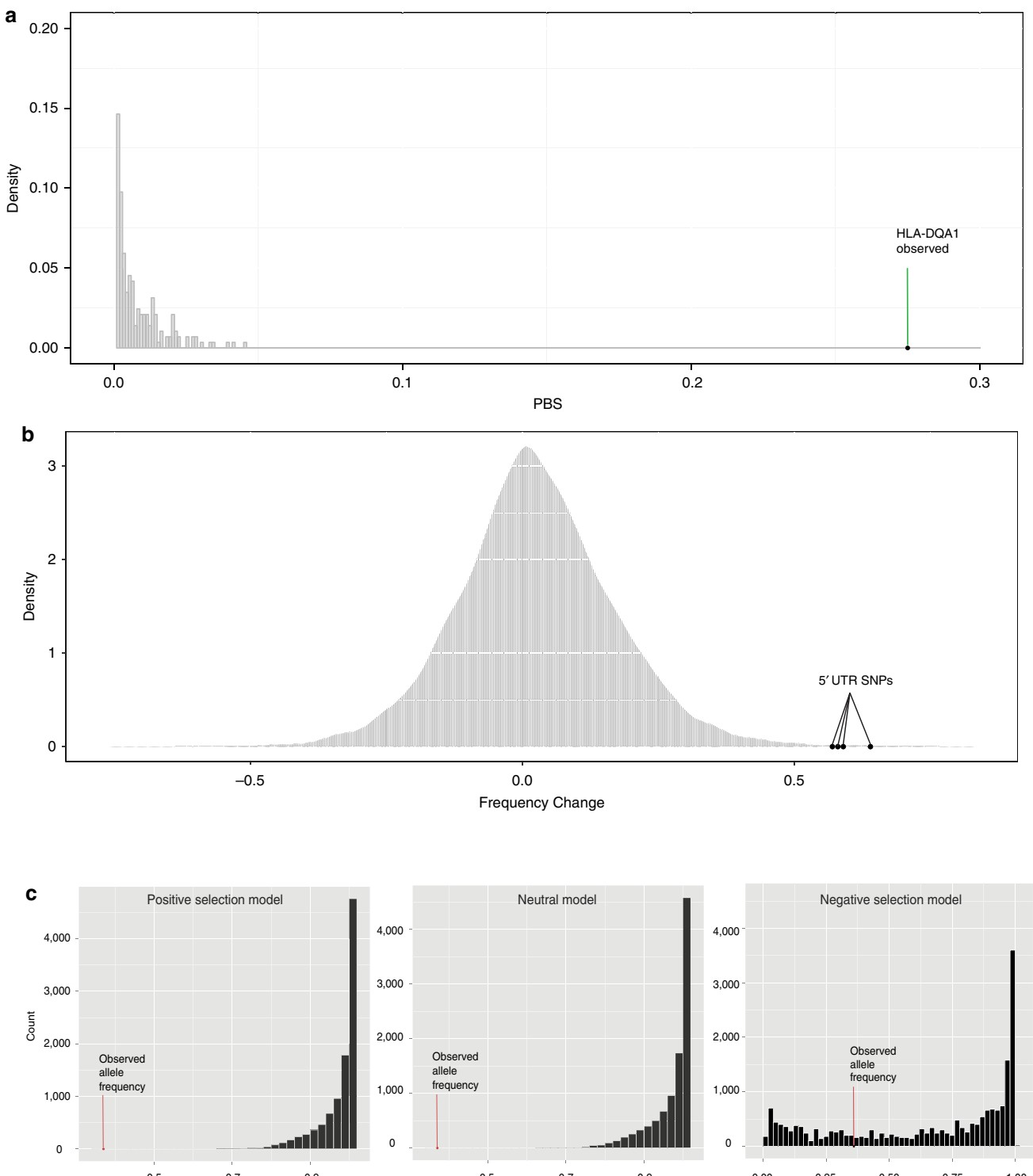

**Figure 4 | Selection scenarios before and after European Contact.** (**a**) Distribution of PBS scores for long-term balancing selection (heterozygote advantage) simulations involving a genomic region of the same length as the *HLA-DQA1* transcript. Given the observed PBS score in the ancient population, this form of balancing selection is inconsistent with our data. (**b**) Empirical distribution of frequency changes for all derived alleles in the exome between the ancient and modern populations, showing the *HLA-DQA1* UTR5 SNPs as outliers. (**c**) Simulations of different selection schemes and the allele frequency of the *HLA-DQA1* SNPs observed in the modern Tsimshian. The allele frequency changes between the ancient and modern populations could not be explained by our demographic model (shift due to neutrality after European contact) or one that included only positive selection (no shift from positive selection after European contact). None of the simulations in which the originally positively-selected allele is neutral or still under positive selection after the time of European contact reached the observed allele frequency in the modern population. A model that incorporated negative selection after European contact was a better fit to our data, where ~26% of the simulations reached the observed frequency.

allele frequency change (since the likelihood surface is relatively flat, Supplementary Fig. 8), it is likely that relatively strong negative selection occurred. Such strength would be expected under a time frame of less than seven generations and correlates with the high mortality rates associated with the regional smallpox epidemics of the 1800s, which reached upwards of 70% (ref. 30).

The results presented here reveal an evolutionary history that spans thousands of years. The immune-related alleles that exhibit strong signals of positive selection in the ancient Native Americans from the Northwest Coast, likely correlate to an adaptation to pathogens that were present in the ancient environments of the region. Our results also suggest that the indigenous population may have experienced negative selection on the same immune-related genetic component after European contact and the ensuing population collapse. The shift may represent a form of balancing selection due to fluctuating environments[49]. This inference was only made possible through our examination of a single population through time, revealing nuanced demographic events and the utility of such studies. Furthermore, the evolutionary history detailed here helps to better understand the experiences of Native Americans with disease, in both ancient and colonial periods, by demonstrating a shift in immune-related selection pressures associated with the environmental impact of European contact.

## Methods

**Ethics and community engagement.** This project was made possible through the active collaboration of the Metlakatla and Lax Kw'alaams First Nations. The communities are located in the Prince Rupert Harbour (PRH) region of British Columbia. RSM and JSC established a collaborative DNA study with these two communities in 2007 and 2008, respectively, visiting annually to report the most recent DNA results and obtain feedback on the results. The 25 exomes from modern individuals generated in this study came from these two communities. The two communities agreed to allow DNA analysis of ancestral individuals recovered from archaeological sites in the region and currently housed at the Canadian Museum of History. During and after community visits and extensive consultation, a research protocol and informed consent documents—agreed on by the indigenous communities and researchers—was approved by the University of Illinois Institutional Review Board (#10538). All individuals signed an informed consent document. RSM, JSC and JL visited the community annually during the study to report the latest results and continue to visit the First Nations to report on this and related studies.

**DNA extraction and library preparation.** We prepared DNA extracts from 25 ancient individuals from the Prince Rupert region of British Columbia (Supplementary Table 1) and prepared DNA sequencing libraries in a clean room facility. The 25 modern DNA samples underwent similar procedures in a separate facility designated for modern DNA only (Supplementary Note 5).

**Exome capture and Illumina sequencing.** For the contemporary samples, a combination of the Illumina TruSeq Exome Enrichment Kit and the Nextera Rapid Capture Exome Kit (Illumina, San Diego, CA) were used (Supplementary Table 2), following the manufacturer's protocol. One library per individual was sequenced (single-end reads) and pooled for a total of four libraries per lane on the Illumina HiSeq 2000 at the High-Throughput Sequencing Division of the W.M. Keck Biotechnology Center at the University of Illinois Urbana-Champaign.

For the ancient samples, only the Illumina TruSeq Exome Enrichment Kit was used. For each ancient individual, four libraries were captured and then pooled for sequencing on one lane. For the capture, the manufacture's protocol was used with the following modifications: the Qiagen MinElute PCR Clean-up kit was used instead of beads, post-capture amplification involved 12 cycles instead of 10 and the hybridization temperature was decreased to 50 °C.

**Contamination estimates.** To estimate contamination across the genome-wide data, we used the ContEst tool[16]. The tool uses a Bayesian approach to calculate both the posterior and the maximum a posteriori probability of contamination level within a BAM file of an individual. This method has been shown effective in detecting contamination in exomes with low coverage[16]. HapMap_3.3 global population frequencies for each SNP, mapped to b37, were used for the estimates. All ancient samples demonstrated contamination below 1%, except for PRH Ancient 163. The estimates are shown on Supplementary Table 3.

**Variant discovery.** See Supplementary Note 6 for details on mapping. Reads below a length of 35 were filtered out before mapping to hg19. For analyses requiring genotype calls (for example, *TreeMix*, *ADMIXTURE*, MDS and *RFMix*),

SAMtools-1.1 (ref. 50) was utilized with a minimum mapping quality of 30, a minimum base quality of 20, a minimum read depth of 6 and a max read depth of 80. Sites were also filtered for violation of a one-tailed test for Hardy–Weinberg Equilibrium at a $P$ value $< 10^{-4}$ (ref. 51). Due to the low-mean read depth of both the PRH Ancients and the Tsimshian, genotypes were not called directly for the selection scan or the demographic modelling. Instead, the program ANGSD (ref. 52) was used to compute genotype likelihoods using the SAMtools model and estimated allele frequencies directly from these likelihoods. This method was applied to all populations considered in the selection scan. For the demographic model, the derived joint site frequency spectra (SFS), for all populations considered, were also inferred using ANGSD. Each alignment used in the estimation was filtered for a minimum mapping quality of 30, a minimum base quality of 20, trimmed at each end for 5 bp to minimize biases from DNA deamination and a minimum $P$ value threshold of $10^{-6}$.

**PBS selection scan.** To detect regions under positive selection in both the PRH Ancient and the Tsimshian, the PBS[31] was utilized. The PBS[31] has proven powerful in detecting hypoxia adaptation in high-altitude populations[31,32]. It uses a set of three populations (call them $X$, $Y$ and $Z$), and assumes that they have the rooted relationship $((X,Y),Z)$. In actuality, the calculation for the PBS does not require a rooted tree, and so their specific rooted relationship does not matter.

An analogous statistic can be calculated for populations $Y$ and $Z$. In this study, we are concerned with the situation in which $X$ is the PRH Ancients, $Y$ is the modern Tsimshian and $Z$ is the Han Chinese (CHB from the 1,000 Genomes Project). We therefore are interested in computing:

$$\text{PBS}_{\text{PRH}} = \frac{T_{\text{PRH,Tsimshian}} + T_{\text{PRH,CHB}} - T_{\text{Tsimshian,CHB}}}{2}.$$

Because the *ADMIXTURE* and *TreeMix* analyses (Fig. 1a,c) indicate a likely admixture event between the modern Tsimshian and Europeans, the allele frequencies in the modern Tsimshian were corrected for admixture using the method described by Huerta-Sánchez et al.[32]. Let $f^*_{\text{Tsimshian}}$ and $f_{\text{Tsimshian}}$ represent the allele frequency at a locus in the Tsimshian population pre and post admixture, respectively. Further, assuming that we use Europeans as a proxy, we let $f_{\text{European}}$ be the allele frequency at the same locus in a reference European population (we used the Great Britain (GBR) population from the 1,000 Genomes Project). Assuming that the proportion of ancestry derived from Europeans at the locus is α, under a model of instantaneous admixture, the allele frequency in the Tsimshian post admixture would be

$$f_{\text{Tsimshian}} = (1 - \alpha)f^*_{\text{Tsimshian}} + \alpha f_{\text{European}}.$$

Rearranging, we can solve for the allele frequency before admixture as

$$f^*_{\text{Tsimshian}} = \frac{f_{\text{Tsimshian}} - \alpha f_{\text{European}}}{1 - \alpha}.$$

We estimated α at each locus by choosing an α value that minimized the $F_{ST}$ between the admixture-corrected Tsimshian and the Han Chinese (CHB) outgroup population. We used the admixture-corrected allele frequencies for the scans for positive selection.

We used ANGSD (ref. 52) to compute allele frequencies for the modern Tsimshian, the PRH Ancient, the Han Chinese (CHB) and the Great Britain (GBR) populations directly from the raw sequencing reads, accounting for the uncertainty in genotype calling. Allele frequencies were based on 25 unrelated individuals (Supplementary Table 10, Supplementary Note 7) from each population. While samples sizes of 10 provide sufficient statistical power for genome-wide $F_{ST}$ differentiation[53], our ancient sample size was increased to 25 (equating to 50 haploid samples) to offset the statistical power loss due to the varying nature of ancient exome coverage. We required that reads had a map quality of at least 30 and each nucleotide had a quality of at least 20. We also only called allele frequencies at sites in which data from at least five individuals was not completely missing. To additionally guard against post-mortem deamination, we trimmed the first and last five nucleotides of each read in the PRH Ancient samples. The allele frequencies in the modern Tsimshian were subsequently corrected for potential European ancestry (see procedure in the directly preceding paragraph).
The total number of loci used (including monomorphic and polymorphic sites) for each scan are as follows: Ancient, Tsimshian, CHB = 2,556,963; Ancient, Tsimshian, CHB (GBR corrected) = 1,594,924; Ancient, Peruvian, CHB = 3,334,664. See Supplementary Note 8 for additional detail on the PBS scan.

**PBS selection scan P values.** To compute $P$ values for the per-gene PBS scan, we first obtained the distribution of RefSeq transcript lengths, and added 20 kilobases to that length. This procedure was to mimic the per-gene scan in which we computed PBS for a given gene with the inclusion of 10 kilobases upstream and downstream of the gene. On the basis of the inferred demographic model (Fig. 2), we performed $10^5$ random neutral simulations using FastSimCoal2 (66). For each replicate simulate, we drew a sequence length uniformly at random from the distribution of RefSeq transcript lengths (plus 20 kilobases). In each simulation, we sampled 50 haplotypes (25 diploid individuals) from each of the four populations (representing PRH Ancient, Tsimshian, CHB and GBR). We then attempted to

correct the allele frequencies in the population representing the modern Tsimshian, using a procedure identical to that described in the 'Methods' section.

$P$ values for the per-gene scan in the PRH Ancients were obtained by identifying the proportion of the $10^5$ neutral simulations in which the PBS values for the population representing the PRH Ancients was more extreme. Associated $P$ values for the top two candidate genes are indicated in Table 2. It should be noted that the $P$ values were generated assuming a neutral model. However, the data are from genes (in particular exomes), which are likely not evolving neutrally and many of which are probably under selective constraint. This selection constraint would act to shift the empirical distribution of PBS values to those that are smaller. Therefore, neutral loci would tend to have higher PBS values. Indeed, contrasting the simulated and empirical PBS distributions (Supplementary Fig. 3), we can see that the empirical distribution of PBS is shifted to smaller values. However, the top candidate HLA-DQA1 is a substantial outlier according to the empirical distribution. Therefore, simulations involving purifying selection rather than neutral simulations would likely have made HLA-DQA1 more significant.

**Demographic history model.** Parameters for the demographic model (Fig. 2) were inferred with FastSimCoal2 (ref. 54). The fixed parameters were implemented from Gravel et al.[55] and were as follows: out of Africa bottleneck ($N = 1,861$, $T = 51$kya)[55], split between the CHB and GBR (serving as the ghost population) ($N_{GBR} = 1,032$, $N_{CHB} = 550$; $T = 23$kya)[55]. Admixture between the GBR and Tsimshian ($T = 100$ years, admixture fraction $= 0.33$) were taken from Verdu et al.[24]. One hundred optimizations were run for the inferred values, taking the best likelihood parameters from each of the 100 sets. The data was simulated with an effective sequence length of 7.4 Mb and per-base per-generation mutation and recombination rates of $2.5 \times 10^{-8}$. The optimizations utilized joint derived SFS for the CHB, PRH Ancients and Tsimshian. The European population (Great Britain denoted by GBR) served as a ghost population in the model. This SFS contained 7.4 Mb of monomorphic and polymorphic sites based on hg19 potential synonymous sites, where data was reported for each individual. A parametric bootstrapping approach was used to construct the 95% confidence intervals. The inferred parameters and confidence intervals are listed in Table 1.

**Long-term balancing simulation under heterozygote advantage.** To examine whether long-term balancing selection could better explain our data than positive selection, the parameters inferred from our demographic model were implemented into the forward-time simulator SLiM (ref. 40). SLiM does not permit continuous exponential growth, so the equivalent effective population size of the CHB and GBR populations were computed. That is, the effective size with the same amount of elapsed coalescent time as one under exponential growth. A de novo mutation was introduced 5 million years in the past (assuming a generation time of 25 years) that evolved under heterozygote advantage (per-generation selection coefficient $s = 0.1$, and dominance parameter $h = 100$) until the present. Only simulations for which the selected mutation was not lost were kept. The simulation involved a region that was equal to the length of the HLA-DQA1 transcript $+ 20$ kb (10 kb upstream and downstream, as in our PBS analysis). Fifty chromosomes were sampled at random in each of the four populations (representing the PRH Ancients, modern Tsimshian, CHB and GBR populations), and PBS was calculated as in all other analyses. An admixture correction was also applied to the data from these simulations. The distribution of the resulting PBS scores can be seen overlapped with the observed data and neutral simulations in Fig. 4a. Because the distribution of the PBS scores under long-term balancing selection is shifted towards small values compared with neutrality, the data are inconsistent with long-term balancing selection.

**Selection shift simulations.** To model evolutionary forces acting on the derived allele at rs9272426 after European contact, we utilized a simulation based approach to evaluate models under positive, neutral and negative selection, similar to that described in Nakagome et al.[56]. We also used the same approach to obtain estimates of the correlation between the time of environmental change ($t$) and the selection coefficient ($s$). First, we ran a Wright–Fisher model based backward simulation of the derived allele frequency in CHB under the neutral model to sample the allele frequency ($f_{T:605}$) at 605 generations ago, which is the estimated time at which ancestral Native Americans split from CHB according to our demographic model, assuming the current frequency was 0.475 (1,000 Genomes CHB frequency) and a constant population size of 8,250 diploid individuals. We computed this constant population size to be the effective size with the same amount of elapsed coalescent time as one under exponential growth assumed in for the CHB in our demographic model. We then started forward simulation in the Tsimshian with the initial frequency of $f_{T:605}$ by taking into account demographic effects based on our model in which effective population size ($N_e$) was 13,975 diploids between 605 and 7 generations ago and decreased to 6,006 diploids at 7 generations ago (Fig. 2). Since we detected signatures of selection on the derived allele based on the PBS statistic in our ancient samples at 60 generations ago ($f_{T:60} = 1.0$), we sampled a selection coefficient ($s_{original}$) from $U(0.0, 0.1)$ and only accepted trajectories if the frequency at 60 generations ago was $> 80$%.

After the trajectory reached 12 generations ago, when European contact occurred in British Columbia, we assumed three different models with or without a shift in selective pressures on this allele by changing $s$ to 0.0 (neutral) or newly sampling $s$ from $U(0.0, -0.3)$ (negative selection) or by using $s = s_{original}$ (positive selection). Then, we calculated the allele frequency at present by binomial sampling with the total chromosomes (50 in our samples) and the current frequency in the trajectory ($f_{present}$).

We estimated a joint posterior distribution of $t$ and $s$ given $f_{T:60}$ in the ancient samples and the observed frequency in the modern samples ($f_{obs}$) under the negative selection model. Similar to the first step, we sampled $f_{T:605}$ from a neutral distribution generated by the backward simulation in CHB and model the trajectory that started from $f_{T:605}$ and increased the frequency with $s_{original}$ sampled from $U(0.0, 0.1)$. The information on $f_{T:60} = 1.0$ was incorporated into the trajectory by rejecting it if $f_{T:60} < 0.8$. At this simulation, we also sampled $t$ from $U(0, 30)$, as well as $s$ from $U(0.0, -0.5)$. We used 0.37 as the observed frequency in our modern Tsimshian sample, with 19 derived and 31 ancestral observed alleles in the modern samples. We estimated a joint posterior distribution of $t$ and $s$ given the observed frequency in our ancient and modern samples by accepting 10,000 samples (Supplementary Fig. 8).

We also investigated if the observed allele frequency in the ancient population could be better explained by drift rather than selection. Using the same general simulation method described above, we simulated the initial allele frequencies at the time Native Americans split off from East Asian populations by randomly sampling allele frequencies from a backward simulation conditioned on the modern CHB frequency. We then simulated allele frequencies at 60 generations ago, the time at which the ancient population was sampled. The resulting distributions in Supplementary Fig. 9 show that a neutral scenario is not a good fit for our data.

**Assessment of differences in coverage between populations.** We investigated the distributions of coverage in the PRH Ancients, modern Tsimshian and CHB populations across the whole genome, across chromosome 6, and in the HLA-DQA1 region (Supplementary Fig. 10). As expected from the degraded nature of ancient DNA, the PRH ancients exhibit more missing data than both modern populations. However, the level of missing data across the genome and across chromosome 6 is similar. The HLA-DQA1 region shows that the coverage for PRH Ancients is less than the modern Tsimshian, and that the modern Tsimshian is less than CHB. Although we observe a decrease in coverage in the modern Tsimshian and PRH ancients relative to the background level of coverage, the number of observed alleles is always $> 20$ (that is, 10 diploid individuals), which is sufficient to compute accurate values of $F_{ST}$ and is over twice as large as the minimal threshold of non-missing individuals (five diploids or 10 alleles) for calling an allele frequency in our ANGSD pipeline. Further, the high frequency variant identified using our PBS scan was confirmed by Sanger sequencing in 18 diploid individuals (36 total alleles) from the PRH Ancients (Supplementary Fig. 4), indicating that it is not sample size that is driving the observed PBS patterns.

**TreeMix analysis.** We started with the identical filtered data set of called genotypes described in the 'Methods' section. TreeMix[26] was applied to the data set to generate maximum likelihood trees and admixture graphs from allele frequency data. The Yoruban (YRI) 1,000 Genomes Project population was used to root the tree (with the –root option). We accounted for linkage disequilibrium by grouping $M$ adjacent sites (with the –$k$ option), and we chose $M$ such that a data set with $L$ sites will have approximately $L/M \approx 20,000$ independent sites. A total of 1,820 polymorphic loci were used for this analysis. At the end of the analysis (that is, number of migrations) we performed a global rearrangement (with the global option). We considered admixture scenarios with $m = 0$ and $m = 1$ migration events. Each migration scenario was run with 100 replicates, and the replicate with the highest likelihood was chosen to represent the maximum likelihood tree or graph for the given migration scenario.

Supplementary Figure 11 displays the results for the maximum likelihood tree with no admixture ($m = 0$) events. Here, the present-day Tsimshian fall ancestral to modern Central and South American samples (Surui, Karitiana and Mayan), as well as the ancient sample from Montana (Anzick-1) and the PRH ancient samples genotyped in this study. However, ADMIXTURE results from Fig. 1a revealed a large European component within the modern Tsimshian, but not in the PRH Ancients, likely causing the modern Tsimshian to fall intermediate between Native Americans and Europeans on the tree. In addition, the Mayan fall ancestral to the Anzick-1 sample, a result of not masking out recent European admixture in that sample (also see Fig. 1c). Figure 1c (TreeMix graph) and Supplementary Fig. 12(b) (corresponding residuals) display results with a single admixture event ($m = 1$). The extra migration event corrects for the European ancestry observed in the modern Tsimshian and shifts the placement on the Tsimshian as a sister population (that is, forms a clade) with the PRH ancient population.

**Data availability.** The ancient data are available from NCBI Sequence Read Archive, accession no. PRJNA288803. The data from modern individuals are

available via a data access agreement with RSM at the University of Illinois. All other data are available from the authors on reasonable request.

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

## Acknowledgements

This project was made possible through the active collaboration of the Lax Kw'alaams and Metlakatla First Nations. We thank Jun Li for furnishing comparative exome data from contemporary Native American populations. We also thank Alvaro Hernandez and Chris Wright at the University of Illinois Biotechnology Center. The research was funded by the National Science Foundation (#DEB-1557151, #BCS-1413551 & #BCS-1518026) and by the Office of the Vice Chancellor of Research, University of Illinois at Urbana-Champaign, by the Canadian Museum of History, Gatineau, Quebec, Canada and by Pennsylvania State University and University of California, Merced startup funds. Portions of this research were performed with the Advanced CyberInfrastructure

computational resources provided by the Institute for CyberScience at Pennsylvania State University.

## Author contributions

Conceived and designed the study R.S.M., J.L. and M.D. Performed the experiments: J.L. and M.R. Analysed the data: J.L., M.D., E.H.-S. and S.N. Contributed reagents/materials/analysis tools: R.S.M., J.L., M.D. and E.W. Wrote the paper: J.L., R.S.M., M.D. and J.S.C. with contributions from all authors. Community engagement: R.S.M., J.S.C., B.P., J.M. and J.L. Discussed and interpreted results: J.L., M.D., R.S.M. and E.H.-S.

## Additional information

**Competing financial interests:** The authors declare no competing financial interests.

