## [Peer review file · Nature Communications]

Reviewers' Comments:

Reviewer #1 (Remarks to the Author)

The manuscript by Lindo et al. reports new exome sequence data from 50 modern and 50 ancient DNA samples from NW coast of North America. As a data resource this work can be seen as a major contribution both in terms of the number of ancient and modern samples covered from North America, plus, also in terms of the depth of sequence coverage of the bulk of ancient samples. The data is used to make inferences about population history and to detect local selection. The analyses revealed continuity between ancient and modern groups, a bottleneck after European contact <200 years ago and a novel selection signal on immunity gene HLA-DQA1. These findings are important in the context of the broader debates about the population history of Native groups in North America, the immune-naivety of Native American populations when confronted with infectious diseases of the Old World, and questions on the adaptation in human populations in general. The main results and key statements are supported by quantitative tests and I do not have any major concerns about the quality of the work.

The concerns of a minor nature that I have are about the presentation.

1. The title may seem to be too complex and difficult to understand for a lay reader, in particular the phrase 'Demographic and immune-based selection shifts'
2. While some detail is provided in the supplementary tables it would be useful for the reader to see some of this information being briefly summarized in the main text. For example, the first paragraph of the results section could summarize the radio-carbon age distribution of the ancient samples to give the reader an idea about how old the samples are.
3. While relative to many previously published ancient genomes the mean depth of 7.97x of the ancient samples may seem high it is still to be considered low for inferring genotypes and distinguishing heterozygotes from homozygotes.
 - a. Unfortunately, not enough detail is provided on how exactly the genotypes were called. For, example, on page 6 it is stated that "To visualize the haplotypes in the HLA-DQA1 region, we phased the ancient and modern samples using Beagle 4.1", which leaves the impression that genotypes were called before phasing could be attempted. On page 11, however, we learn that "Due to the low average read depth of the PRH Ancients, genotypes were not called directly for the selection scan."
 - b. It is quite unclear here why the average coverage of 7.97x in ancient samples was deemed to be that much lower for the purpose of genotype calling than the 9.66x in the modern samples. Both these depths are to be considered much lower than the level of 'high coverage' which is normally used for calling genotypes.
 - c. It should be also noticed that while the average read depth values for the modern samples are reported in Table S2, it would be helpful if the individual depth values for the ancient samples were also reported, e.g. in Table S3 as an additional column.
4. The presentation of the PBS results is not sufficiently reader-friendly. When going back and forth between the main text/tables and the supplementary material one can find out the details but it would be helpful if relevant main text paragraphs/table footnotes/figure labels had sufficient clarity on whether particular findings that are presented were detected on a SNP-wise or gene-wise approach, what was the general basis by which the reported p values were calculated, what was the cut-off for the presented gene lists.

Reviewer #2 (Remarks to the Author)

Review: Lindo et al 2016 NCOMMS-16-10030

In this study, Lindo et al. investigated the potential impact of infectious disease introduced by Europeans on the genetic make-up of ancient and modern Tsimshian populations. In particular, they were interested in determining whether a preexisting genetic component in Tsimshian

individuals led to their susceptibility to these diseases. Through an exome analysis, they showed that there was some genetic continuity between the ancient and modern individuals, and observed a significant reduction in effective population size following European contact. Interestingly, they found evidence for positive selection in ancient but not modern Tsimshian individuals, suggesting an ancient adaptation to local pathogens, with the locus showing the strongest signal being HLA-DQA1. The frequencies of the alleles associated with this response were lower in the modern Tsimshian, probably due to negative selection on this locus because of exposure to infectious diseases introduced by Europeans. Through this analysis, they further developed a more detailed demographic history of the Tsimshian, including evidence of European admixture in the 19th century following disease epidemics.

This is a very interesting study of immunogenetic variation and selection in a First Nations population from British Columbia. It has leveraged information from ancient human samples to assess the impact of infectious disease on indigenous peoples and also the effects of selection on these populations through their adaptation to local environments prior to European contact. How this kind of adaptation took form is the focus of my questions below.

The authors used a variety of approaches to evaluate the relatedness of the ancient and modern samples, and model the demographic history of the Tsimshian. The simulation of the Tsimshian bottleneck was particularly intriguing given historical records for population crashes across northern North America in the 19th century. Through the PBS selection scan, they identified loci involved in immune function as having the strongest signals of positive selection, a finding that is generally consistent with other studies of ancient DNA and infectious disease, such as tuberculosis. The effort to delineate the possible effects of balancing or negative selection and drift using SLiM were also illuminating. Moreover, the diachronic perspective afforded by the analysis of ancient and modern DNA samples is also one that is often lacking in studies of disease and selection in contemporaneous groups.

While impressed with the sophistication of this analysis and the results of the study, I do have some specific concerns about the study that I have outlined below.

Comments and Questions

Page 2, Line 44: What might the local pathogens to which the ancient Tsimshian were adapted prior to European contact? What can be surmised about these pathogens based on the genes identified as possibly being involved in the selective events shaping variation at the HLA-DQA1 locus (Supplemental Tables 4-6)?

Page 2, Line 62: While describing decontamination methods in the supplemental materials, the authors might simply mention here that "contamination" in this sentence means the presence of modern DNA in ancient samples.

Page 2, Lines 66-67: All PRH Ancients had mtDNA haplotypes previously observed in modern Native Americans. What does this finding mean in relation to recent studies of ancient mitogenomes that have shown no overlap between the haplotypes from modern and ancient populations of the same regions (e.g., Llamas et al. 2016)?

Page 2, Lines 70-73: To be clear here, ALL of the C/T and G/A variants were removed from the exomic sequences obtained from the PRH Ancients and modern Tsimshian samples prior to sequence analysis, correct? Given that the vast majority of mutations in the genome are such transitions, how many polymorphic sites were left for comparison?

Page 3, Lines 80-85: The evidence for genetic continuity between ancient and modern Tsimshian samples is generally compelling, although the dates for the ancient PRH samples span some 6,000 years of time. The fact that the ancient individuals are not related to each other (Supplemental Table 11) reflects the temporal distance between the 25 different samples. While this fact

eliminates the issue of kinship influencing patterns of genetic variation in the ancient samples, it also raise the question as to how accurately one can model selective forces on such an ancient "population" given the time range that the samples represent. Will selection on the immune genes under study have been constant over this 6,000-year time period? What constitutes a "local environment" from the standpoint of selection here?

This geographic region of North America is also known to be fairly dynamic in terms of population movements between the interior and coastal areas of the region, particularly over the past two thousand years. Thus, the Tsimshian should probably not be viewed as an isolated group living in the Prince Rupert Harbor area for the entire 6,000-year time span represented by the ancient samples. In light of this demographic history, does the signal of positive selection at the HLA-DQA1 reflect a specific adaptation to local pathogens or a more general immunological response shared by Native Americans as a result of adaptation to novel environments during the initial colonization and later settlement of the Americas?

Suggested Revisions

1. The labels appearing in Figure 2 should be defined in the caption for this figure.
2. Supplemental Figure 3 does not seem to have a figure caption, unless it is supposed to a continuation of Supplemental Figure 1. If this is the case, then the title needs to be renamed.
3. Supplemental Figure 4 is missing by my reckoning.
4. Supplemental Figure 8 legend: "The HLA-DQA1 region shows that the coverage for PRH Ancients is less than THAT FOR the modern Tsimshian, and that the modern Tsimshian HAVE LESS COVERAGE than CHB."
5. Supplemental Figure 15 is difficult to read carefully, given its reduction in size in the submission process. This image needs to be made as large as possible in the journal to allow readers to get a glimpse of the two haplotypes for PRH 125.
6. I suggest that the authors number the supplemental tables and figures in the order that they are cited in the text of the paper. This is not done consistent in the submitted manuscript.
7. Check the wording of all supplemental figure captions for clarity.

Reviewer 1 comments:

The manuscript by Lindo et al. reports new exome sequence data from 50 modern and 50 ancient DNA samples from NW coast of North America. As a data resource this work can be seen as a major contribution both in terms of the number of ancient and modern samples covered from North America, plus, also in terms of the depth of sequence coverage of the bulk of ancient samples. The data is used to make inferences about population history and to detect local selection. The analyses revealed continuity between ancient and modern groups, a bottleneck after European contact <200 years ago and a novel selection signal on immunity gene HLA-DQA1. These findings are important in the context of the broader debates about the population history of Native groups in North American, the immune-naivety of Native American populations when confronted with infectious diseases of the Old World, and questions on the adaptation in human populations in general. The main results and key statements are supported by quantitative tests and I do not have any major concerns about the quality of the work. The concerns of a minor nature that I have are about the presentation.

We appreciate the reviewer's acknowledgement of both the importance of our research question as well as the novelty of the dataset.

1. The title may seem to be too complex and difficult to understand for a lay reader, in particular the phrase 'Demographic and immune-based selection shifts'

We have changed the title to "A time transect of exomes from a Native American population before and after European contact", so that it is more easily understood by a lay reader.

2. While some detail is provided in the supplementary tables it would be useful for the reader to see some of this information being briefly summarized in the main text. For example, the first paragraph of the results section could summarize the radio-carbon age distribution of the ancient samples to give the reader an idea about how old the samples are.

The first paragraph of the results section (page 3, line 67) now summarizes the findings presented in Supplementary Tables 1, 2, and 3, which include a date range of the ancient individuals, the mean sequencing read depth, a range of contamination estimates, and the mitochondrial haplogroups.

3. While relative to many previously published ancient genomes the mean depth of 7.97x of the ancient samples may seem high it is still to be considered low for inferring genotypes and distinguishing heterozygotes from homozygotes.

We appreciate the concern raised by the reviewer with the relatively low sequencing read depth achieved in the ancient samples to call genotypes. In order to avoid potential biases from the low coverage data in the selection scans, we did not call genotypes. Instead, we estimated allele frequencies for the ancient population using the ANGSD software. The method to estimate allele frequency in ANGSD allowed us to take into account the uncertainty in genotype calling by circumventing the genotype calling stage and instead inferring sample allele frequencies directly from genotype likelihoods, and has been shown to outperform genotype calling methods in low coverage data.¹ We describe our method in greater detail in the Methods under "Variant discovery" (page 12, line 363).

¹Kim, S. Y., Lohmueller, K. E., Albrechtsen, A., & Li, Y. (2011). Estimation of allele frequency and association mapping using next-generation sequencing data. *BMC Bioinformatics*. 12:231.

a. Unfortunately, not enough detail is provided on how exactly the genotypes were called. For, example, on page 6 it is stated that "To visualize the haplotypes in the HLA-DQA1 region, we phased the ancient and modern samples using Beagle 4.1", which leaves the impression that genotypes were called before phasing could be attempted. On page 11, however, we learn that "Due to the low average read depth of the PRH Ancients, genotypes were not called directly for the selection scan."

The reviewer is correct that for certain analyses we called genotypes in the ancient and modern individuals. However, we did so only in cases of visualization or where locus specific calls were not crucial to the analysis. These additional analyses involved *ADMIXTURE*, *TreeMix*, *RFMix*, and haplotype visualization. For these analyses, reads were filtered for quality, DNA damage, and read depth. These analyses have previously been used in ancient DNA studies with genotype calls despite low coverage since the bias introduced from such calls is outweighed by the overall ancestral information gained across loci.

We have now added more detail to the Methods section under "Variant discovery," (page 12, line 363) which details the analyses that included genotype calls.

b. It is quite unclear here why the average coverage of 7.97x in ancient samples was deemed to be that much lower for the purpose of genotype calling than the 9.66x in the modern samples. Both these depths are to be considered much lower than the level of 'high coverage' which is normally used for calling genotypes.

The reviewer is correct that both the modern and the ancient samples should be considered low coverage for purposes of genotype calling. In light of this, we performed the same procedure to estimate allele frequencies for the selection scans and the derived site frequency spectra for the inference of the demographic model, for both the modern and ancient populations (page 12, lines 367-373).

c. It should be also noticed that while the average read depth values for the modern samples are reported in Table S2, it would be helpful if the individual depth values for the ancient samples were also reported, e.g. in Table S3 as an additional column.

We appreciate the reviewer's suggestion and have now added the ancient and modern read depth to Supplementary Table 2.

4. The presentation of the PBS results is not sufficiently reader-friendly. When going back and forth between the main text/tables and the supplementary material one can find out the details but it would be helpful if relevant main text paragraphs/table footnotes/figure labels had sufficient clarity on whether particular findings that are presented were detected on a SNP-wise or gene-wise approach, what was the general basis by which the reported p values were calculated, what was the cut-off for the presented gene lists.

We agree with the reviewer that our original presentation of the PBS results was not sufficiently clear. We have now modified the Results section under "Scans for positive selection" (page 5, line 148), to allow the reader to grasp our method without needing to refer to the supplement.

Reviewer 2 Comments:

In this study, Lindo et al. investigated the potential impact of infectious disease introduced by Europeans on the genetic make-up of ancient and modern Tsimshian populations. In particular, they were interested in determining whether a preexisting genetic component in Tsimshian individuals led to their susceptibility to these diseases. Through an exome analysis, they showed that there was some genetic continuity between the ancient and modern individuals, and observed a significant reduction in effective population size following European contact. Interestingly, they found evidence for positive selection in ancient but not modern Tsimshian individuals, suggesting an ancient adaptation to local pathogens, with the locus showing the strongest signal being HLA-DQA1. The frequencies of the alleles associated with this response were lower in the modern Tsimshian, probably due to negative selection on this locus because of exposure to infectious diseases introduced by Europeans. Through this analysis, they further developed a more detailed demographic history of the Tsimshian, including evidence of European admixture in the 19th century following disease epidemics.

This is a very interesting study of immunogenetic variation and selection in a First Nations population from British Columbia. It has leveraged information from ancient human samples to assess the impact of infectious disease on indigenous peoples and also the effects of selection on these populations through their adaptation to local environments prior to European contact. How this kind of adaptation took form is the focus of my questions below.

The authors used a variety of approaches to evaluate the relatedness of the ancient and modern samples, and model the demographic history of the Tsimshian. The simulation of the Tsimshian bottleneck was particularly intriguing given historical records for population crashes across northern North America in the 19th century. Through the PBS selection scan, they identified loci involved in immune function as having the strongest signals of positive selection, a finding that is generally consistent with other studies of ancient DNA and infectious disease, such as tuberculosis. The effort to delineate the possible effects of balancing or negative selection and drift using SLiM were also illuminating. Moreover, the diachronic perspective afforded by the analysis of ancient and modern DNA samples is also one that is often lacking in studies of disease and selection in contemporaneous groups.

While impressed with the sophistication of this analysis and the results of the study, I do have some specific concerns about the study that I have outlined below.

We appreciate that the reviewer finds our manuscript interesting and important.

Comments and Questions

Page 2, Line 44: What might the local pathogens to which the ancient Tsimshian were adapted prior to European contact? What can be surmised about these pathogens based on the genes identified as possibly being involved in the selective events shaping variation at the HLA-DQA1 locus (Supplemental Tables 4-6)?

The reviewer poses an interesting question. Although we would need additional evidence from sources such as pathogens extracted from ancient individuals in the region to investigate this in a future study, we do know that *HLA-DQ* interacts with antigens from a broad range of pathogens, including viruses and bacteria. Interestingly, alleles of *HLA-DQ* genes have been associated with a variety of colonization era infectious diseases, including measles^{1,2}, tuberculosis^{3,4}, and with the adaptive immune response to the vaccinia virus, which is an attenuated form of smallpox^{5,6}. We do know that the variation between the ancient and modern is based in the UTR5 region of the gene, which suggests that selection occurred on the regulation of the receptor. The potential alteration in how often the receptor was expressed may have conveyed an adaptive advantage to a pathogen, or a class of pathogens, in the ancient environments of the Americas.

¹Ovsyannikova IG, Vierkant RA, Poland GA (2006) Importance of HLA-DQ and HLA-DP polymorphisms in cytokine responses to naturally processed HLA-DR-derived measles virus peptides. *Vaccine* 24 (25):5381–5389.

²Moss WJ, Griffin DE (2006) Global measles elimination. *Nat Rev Micro* 4(12):900–908.

³Kim HS, et al. (2005) Association of HLA-DR and HLA-DQ genes with susceptibility to pulmonary tuberculosis in Koreans: preliminary evidence of associations with drug resistance, disease severity, and disease recurrence. *Hum Immunol* 66(10):1074–1081.

⁴Delgado JC, Baena A, Thim S, Goldfeld AE (2006) Aspartic acid homozygosity at codon 57 of HLA-DQ beta is associated with susceptibility to pulmonary tuberculosis in Cambodia. *The Journal of Immunology* 176(2):1090–1097.

⁵Ovsyannikova IG, Vierkant RA, Pankratz VS, Jacobson RM, Poland GA (2011) Human Leukocyte Antigen Genotypes in the Genetic Control of Adaptive Immune Responses to Smallpox Vaccine. *Journal of Infectious Diseases* 203(11):1546–1555.

⁶Ovsyannikova IG, Pankratz VS, Salk HM, Kennedy RB, Poland GA (2014) HLA alleles associated with the adaptive immune response to smallpox vaccine: a replication study. *Hum Genet* 133(9):1083–1092.

Page 2, Line 62: While describing decontamination methods in the supplemental materials, the authors might simply mention here that "contamination" in this sentence means the presence of modern DNA in ancient samples.

We have added a statement under the Results heading: “Samples and sequencing,” clarifying the meaning of contamination in the context of the paper (page 3, line 77).

Page 2, Lines 66-67: All PRH Ancients had mtDNA haplotypes previously observed in modern Native Americans. What does this finding mean in relation to recent studies of ancient mitogenomes that have shown no overlap between the haplotypes from modern and ancient populations of the same regions (e.g., Llamas et al. 2016)?

The reviewer raises an interesting point regarding mtDNA haplotypes before European contact. Unfortunately, the exome capture only resulted in partial sequences from the mitochondria. The resulting mtDNA haplogroup assignments are based on partial mitogenomes or from restriction fragment length polymorphism (Supplementary Table 1), which is not comparable to the complete mitogenomes results from Llamas et al. (2016). However, of the few complete mitogenomes from the same region analyzed in Cui et al. (2013), we did find one shared mitogenome between an ancient individual at 2500 YBP and a living individual from this same region.

Cui, Y. *et al.* Ancient DNA Analysis of Mid-Holocene Individuals from the Northwest Coast of North America Reveals Different Evolutionary Paths for Mitogenomes. *PLoS One* 8, e66948 (2013).

Page 2, Lines 70-73: To be clear here, ALL of the C/T and G/A variants were removed from the exomic sequences obtained from the PRH Ancients and modern Tsimshian samples prior to sequence analysis, correct? Given that the vast majority of mutations in the genome are such transitions, how many polymorphic sites were left for comparison?

We now include details on how many sites were used after filtering in the method sections of the main text (page 13, line 417) and supplement (supplement page 26). The additional information is as follows:

ADMIXTURE: 29,333 polymorphic loci
Multidimensional scaling: 29,333 polymorphic loci
TreeMix: 1,820 polymorphic loci

PBS scans (includes both polymorphic and monomorphic sites):

Ancient, Tsimshian, CHB = 2,556,963 loci

Ancient, Tsimshian, CHB (GBR corrected) = 1,594,924 loci

Ancient, Peruvian, CHB = 3,334,664 loci

Page 3, Lines 80-85: The evidence for genetic continuity between ancient and modern Tsimshian samples is generally compelling, although the dates for the ancient PRH samples span some 6,000 years of time. The fact that the ancient individuals are not related to each other (Supplemental Table 11) reflects the temporal distance between the 25 different samples. While this fact eliminates the issue of kinship influencing patterns of genetic variation in the ancient samples, it also raises the question as to how accurately one can model selective forces on such an ancient "population" given the time range that the samples represent. Will selection on the immune genes under study have been constant over this 6,000-year time period? What constitutes a "local environment" from the standpoint of selection here?

The reviewer raises another interesting set of questions. Given that we have a time series of samples, ranging some 6,000 years, and see evidence that most of the alleles in the UTR5 region are close to fixation, we could infer a couple of scenarios. First, it is possible that a near selective sweep occurred before the first sampling point, whereby alleles in the ancestral population were driven to high frequency. It is also possible that purifying selection was subsequently active, keeping the alleles in high frequency through the last time point of about 1000 years cal BP. Of course, this is all dependent as to when the selective pressure, possibly by a pathogen, was exerted on the population. This then complicates the concept of "local," with regard to selection. If the ancestral population experienced selection before arriving to the region, then the event may not be local to British Columbia but perhaps to a much larger area. On the other hand, if selective pressures were indeed continuous in the region for some 5,000 years, then the event could be considered local adaptation. This is another area where metagenomics and the extraction of ancient pathogens from human remains could be useful for future work in elucidating past adaptive scenarios.

This geographic region of North America is also known to be fairly dynamic in terms of population movements between the interior and coastal areas of the region, particularly over the past two thousand years. Thus, the Tsimshian should probably not be viewed as an isolated group living in the Prince Rupert Harbor area for the entire 6,000-year time span represented by the ancient samples. In light of this demographic history, does the signal of positive selection at the HLA-DQA1 reflect a specific adaptation to local pathogens or a more general immunological response shared by Native Americans as a result of adaptation to novel environments during the initial colonization and later settlement of the Americas?

This is yet another interesting question. Unfortunately, there is little population level whole genome data on Native Americans that we could use to test if an immune adaptation was a widespread event. However, we examined the genome of Anzick-1, a 12,000-year-old individual found in Montana that was sequenced to 14x coverage. The individual only shared 4 of the 9 high frequency SNPs shown on Table 4, suggesting that the selection event did not include the ancestral population linked to Anzick-1.

Suggested Revisions

1. The labels appearing in Figure 2 should be defined in the caption for this figure.

We have revised the caption to include the label definitions (page 5, line 133).

2. Supplemental Figure 3 does not seem to have a figure caption, unless it is supposed to a continuation of Supplemental Figure 1. If this is the case, then the title needs to be renamed.

We thank the reviewer for pointing this out, and it is now corrected (Supplementary Fig. 2, supplement page 6).

3. Supplemental Figure 4 is missing by my reckoning.

We again thank the reviewer for pointing out this typo, which is now fixed.

4. Supplemental Figure 8 legend: "The HLA-DQA1 region shows that the coverage for PRH Ancients is less than THAT FOR the modern Tsimshian, and that the modern Tsimshian HAVE LESS COVERAGE than CHB."

We have modified the figure caption (Supplementary Fig. 13, supplement page 14).

5. Supplemental Figure 15 is difficult to read carefully, given its reduction in size in the submission process. This image needs to be made as large as possible in the journal to allow readers to get a glimpse of the two haplotypes for PRH 125.

We have now enlarged the haplotype graph, which now encompasses an entire page (Supplementary Fig. 6, supplement page 9).

6. I suggest that the authors number the supplemental tables and figures in the order that they are cited in the text of the paper. This is not done consistent in the submitted manuscript.

We now number the tables and figures based on the order that they are cited in the text.

7. Check the wording of all supplemental figure captions for clarity.

We have double checked the wording for clarity.

Reviewers' Comments:

Reviewer #1 (Remarks to the Author)

Reviewer #2 (Remarks to the Author)

The authors have largely addressed my primary concerns about the analysis of ancient and modern Tsimshian samples. The minor editorial and presentation concerns have also been taken care of.

I think the issue of population continuity in the region is still a bit tricky, but their remarks in the rebuttal indicate that they are thinking very carefully about this issue and its genetic implications.